# European Union Legislation for the Welfare of Animals Used for Scientific Purposes: Areas Identified for Further Discussion

**DOI:** 10.3390/ani13142367

**Published:** 2023-07-20

**Authors:** Katerina A. Marinou, Ismene A. Dontas

**Affiliations:** 1Directorate of Animal Welfare, Veterinary Medicines and Veterinary Applications, General Directorate of Veterinary Services, Ministry of Rural Development and Food, 2 Acharnon Street, 10176 Athens, Greece; 2Laboratory for Research of the Musculoskeletal System, School of Medicine, National and Kapodistrian University of Athens, KAT General Hospital, 14561 Kifissia, Greece

**Keywords:** animal welfare, authorization, education, legislation, project, report, severity

## Abstract

**Simple Summary:**

The Directive 2010/63/EU has been in force for more than a decade and has brought about significant improvements and novel requirements for the welfare of animals used for scientific purposes, while safeguarding research integrity. The Directive sets clear provisions for its scope, the authorization procedures, animal welfare bodies, national committees, cost/benefit analysis, severity classification, retrospective assessment, and statistical reporting, among many others. From this perspective, indicative areas in the existing legislative texts and guidance documents, which may influence animal welfare and scientific integrity, have been identified by the authors. Suggested solutions to address these areas by potential future revisions in the legislation are discussed, with the aim to clarify and simplify them for all actors involved.

**Abstract:**

The Directive 2010/63/EU of the European Parliament and of the Council has been in force for more than a decade and has brought about significant improvements and novel requirements for the welfare of animals used for scientific purposes, while safeguarding research integrity. The Directive sets clear provisions for its scope, the authorization procedures, animal welfare bodies, national committees, cost/benefit analysis, severity classification, retrospective assessment, and statistical reporting, among many others. From this perspective, indicative areas in the existing legislative texts and guidance documents, which may influence animal welfare and scientific integrity, have been identified by the authors after years of working in this field. Suggested solutions to address these areas by potential future revisions in the legislation or other actions to benefit animal welfare are discussed, with the aim to clarify and simplify them for all stakeholders involved.

## 1. Introduction

The Directive 2010/63/EU is the outcome of a decade of consultation among stakeholders in order to uphold animal welfare, while concomitantly enabling scientists to conduct research under specific regulations [1]. The Directive 86/609/EC [2] was in place for more than 20 years and, although it had many positive aspects, for example, protecting animal welfare and imposing restrictions on the educational background of persons using animals for scientific purposes, there was a clear need for an update.

The revision, which led to the Directive 2010/63/EU, was based on four main sectors, including scope, authorization procedures, ethical review process, and cost/benefit analysis. Furthermore, significant issues such as education and training of staff, national committees for the protection of animals used for scientific purposes, alternative approaches, new requirements for social housing, and provision of enrichment to animal enclosures were included. Regulated procedures were defined as those that are at least equivalent to the insertion of a needle in accordance with good veterinary practice.

The Directive 2010/63/EU currently provides a high welfare standard for animals used for scientific purposes. Furthermore, the amendment of the Directive with Regulation 2019/1010/EU contributed significantly to the increase in the transparency of information for the public, since the latter imposes horizontal provisions on the alignment of reporting obligations in the field of environmental legislation “and increasing transparency for the benefit of the public, for instance with the use of harmonized templates for the publication on non-technical summaries, related retrospective assessments as well as implementation and statistics” [3]. Moreover, a large body of recommendations and guidelines has been produced for the same purpose of ensuring animal welfare and scientific integrity by the European Commission [4], as well as national bodies, non-governmental organizations, and scientific associations.

The aim of the present perspective is to highlight areas that can be seen as grey zones that may affect animal welfare and scientific integrity in the existing legislative framework and guidance documents, as well as to indicate some gaps and issues identified in practice during recent years in Greece that could be addressed by potential revisions. This perspective is not intended to come across as overly critical, but aims very much at trying to be beneficial and to have a positive impact on the application of the Directive and animal welfare.

## 2. Main Observations and Discussion

In the following sections, indicative areas in the legislative or guidance documents, identified as having a potential effect on animal welfare with possibilities for improvement, have been selected based on our experience with the issues that have arisen during the implementation of Directive 2010/63/EU in Greece, without this selection being exhaustive. While we acknowledge that our experience with the Directive derives from working in one Member State, several of these issues may also potentially be applicable to other Member States to a certain degree. The areas of concern are briefly described and discussed, in the sequence that they are presented in the Directive, and future solutions or actions to be taken are suggested.

### 2.1. Education and Training

The Education and Training (E&T) Framework document developed by the European Commission, as guidance to assist the Member States on how the requirements of the Directive may be met, and to facilitate its implementation, sets out the knowledge and skills required to ensure competence and facilitate free movement of personnel [5]. It is recommended within it that enrollment in training courses for Functions A, C, and D (persons carrying out procedures, taking care of animals, and killing animals, respectively), should not require previous specific educational qualifications. Namely, when trainees attend training modules that fulfill the agreed quality criteria set and are assessed in a consistent manner, passing the required modules successfully infers a level of schooling and maturity sufficient for these (three) Functions. The qualifications however, recommended to enroll in a Function B (persons designing procedures and projects) course differ: the Framework document states that these people should normally hold an academic degree or equivalent in an appropriate scientific discipline. The latter recommendation of “equivalent” may be seen as a grey zone regarding the qualifications that are required to enroll in a Function B course, which may differ between Member States, depending on how the Directive has been transposed into each State’s national legislation. The qualifications could be interpreted as having attended brief/extended seminars or webinars, or having a certification of experience issued from an academic department. This potentially wide variety in starting qualifications can hinder the harmonization of E&T and the free movement of persons designing procedures and projects. A potential interpretation according to the E&T Framework document may be that undergraduate students—even from veterinary or biology departments—might not be accepted to participate in a Function B course in some Member States before their graduation, regardless of their current expertise. It is therefore suggested that the starting qualifications of persons wishing to attend a course on designing procedures and projects are clarified in a future legislative revision, so that a uniform approach is applied in the EU Member States.

Member States may of course set strict educational requirements in their transposition of the Directive to safeguard animal welfare.

Additionally, it may be beneficial to animal welfare if there were many available courses to train official inspectors in their duties, who work for the Competent Authorities and conduct audits of establishments, i.e., official controls for the implementation of the Directive 2010/63/EU in breeder, supplier and user establishments. A possible solution could be for the European Commission to organize training courses similar to those of DG SANTE “Better Training for Safer Food” (BTSF), which is a training initiative for Competent Authorities and stakeholders in Member States and non-EU countries, for various topics including animal welfare official controls [6].

### 2.2. Animals Used in Higher Education and Training, or Training for the Acquisition, Maintenance, or Improvement in Vocational Skills

This is an important topic regarding the purposes for which procedures on live animals may be carried out, as mentioned in element (f) of Article 5 of the Directive 2010/63/EU [1]. As a specific example for the training of veterinary students, the European Association of Establishments for Veterinary Education (EAEVE) [7] manages the European System of Evaluation of Veterinary Training (ESEVT) [8], in collaboration with the Federation of Veterinarians of Europe. Many EAEVE accreditation reports state the need for the use of non-animal alternatives in training in veterinary schools. In physiology and pharmacology departments, live animals are not allowed to be used during training exercises of students. In addition, training of veterinary students for routine techniques, such as blood sampling in healthy dogs and cats, has been performed either in animals admitted to small animal clinics for diagnosis and treatment, or in healthy dogs kept by the veterinary school as blood donors, until recently. The same might apply for farm animals. Requirements to replace these live animals with dummies are recorded [8]; however, arguments may be raised especially concerning the cost of their acquisition and maintenance, and the number of students that can be trained per device.

In addition to veterinary students, scientists using animals for scientific purposes also need to be trained using healthy live animals for some procedures in order to acquire the necessary skills and become competent in the conduct of these procedures. As a general conclusion, education and training needs, particularly of some scientific professions whose hands-on learning components on live animals are basic for the acquisition of competence and expertise, should be better defined, indicating, where appropriate, a tier-level approach, e.g., from dummy alternatives to dead animals to anesthetized animals to awake animals [5,9]. Additionally, it is suggested that clarifications could be presented regarding possible exceptions to training on live animals that could be applied in view of the Replacement principle.

### 2.3. Project Evaluation

The process of project evaluation as described in article 38 of the Directive 2010/63/EU requests the participation of experts in the required fields, but also in the process itself, e.g., the evaluation of application forms according to legal provisions, the application of an ethical evaluation, and a harm/benefit analysis of the use of animals for the approval of a project. In most Member States, there are courses that are developed for Functions A, B, C, and D, designed according to their description in article 23 of Directive 2010/63/EU and in the E&T Framework document [1,5]. The E&T Framework document includes a project evaluator module (no. 25); however, to the best of our knowledge, it is currently delivered only in Spain and not in many Member States, as are the other courses for the four Functions. Indicatively, such course listings can be found on the E&T Platform for Laboratory Animal Science (ETPLAS) website (https://courses.etplas.eu/en, accessed on 30 June 2023) [10] and on the Federation of Laboratory Animal Science Associations (FELASA) website (https://felasa.eu/education-training/course-listings, accessed on 30 June 2023) [11]. The provision of a project evaluators’ course with common training content in all Member States would guarantee a harmonized education and training of the members of project evaluation committees and, therefore, a harmonized approach to all project applications, resulting also in free movement of personnel in the European Union. Furthermore, the local module no. 50 of the E&T Framework document offers deeper knowledge about the local environment for persons taking specific roles under the Directive with only a few learning outcomes, which is also important for project evaluators. The proposed solution to this issue could be the development either of courses held in the Member States with a physical presence, or of e-learning modules that could be accessed by project evaluation committee members, such as the one freely available on the ETPLAS website [10].

### 2.4. Animal Welfare Bodies

The composition of the Animal Welfare Bodies (AWBs) in most Member States is defined by the Directive and its transposition to national legislation. In practice, the diversity of AWB’s membership depends on the availability of specialties within each establishment. The input of the designated Veterinarian is required by the Directive, without defining a mandatory participation amongst the AWB’s members. Many Member States have incorporated this designated Veterinarian membership requirement into their national legislation [12], which we believe is to the benefit of animal welfare and should be required in a future legislative revision. Furthermore, it would also be beneficial to the animals if Animal Welfare Body members collaborating with different establishments also participated in their composition as suggested [12].

### 2.5. Quality of the Results of Retrospective Assessment

Non-technical summaries of projects with severe procedures, the use of non-human primates, or for other reasons justified by the evaluating Competent Authority (e.g., pilot projects) need to be updated by the latter with the results of a retrospective assessment report according to article 43 (2) of the Directive 2010/63/EU as amended by Regulation 2019/1010/EU [1,3]. In order to enhance harmonization of data presented to the public, these have been published in a special European Commission database webpage since 2021 [13]. To date, Competent Authorities have been performing this task successfully, despite the occasional difficulty in some cases/countries in retrieving data on actual severity and harm/benefit analysis by the researchers, who might not work for the user establishment at the time of data collection. The response rate of some countries’ Competent Authorities could also be low, due to possible work overload and lack of sufficient personnel. The guidance document provided by the European Commission assists both researchers, project evaluation committee members, and the Competent Authorities in providing and collecting the necessary information [14]. We suggest that, in future legislative documents, binding provisions for the submission of necessary documentations by the users within clear deadlines should be imposed. Furthermore, Competent Authorities themselves should dedicate more specialized staff in evaluating these data, producing high-quality assessments that could fulfill their role in improving science and avoiding duplications in various countries. Finally, in some Member States there is currently internal discussion between authorizing bodies and central Competent Authorities (when different) about whether pilot projects should also de facto require retrospective assessment, which we propose could be defined, at least for some cases, in future legislation.

### 2.6. Quality of Non-Technical Project Summaries (NTSs)

NTSs that are produced in EU Member States have been published in the NTS European Commission open webpage [13] since 1/1/2021 in order to increase transparency and access to knowledge, and for avoidance of duplicated authorized protocols according to the provisions of Regulation 2019/1010/EU [3]. Before 2021, where NTSs were published in national open access webpages (that usually belonged to the Competent Authorities), it was clear that there was still a significant variation in the content of the NTSs in the different fields of the templates among researchers in the same Member State, as well as between different Member States. This situation caused a discrepancy in the data presented to the public, since in many cases the real impact of procedures on animals is not clearly described. Furthermore, it has been shown that the section offering explanations for why the existing alternative methods cannot be used for the purposes of the project is inadequately presented in many cases. The European Commission, in order to address the above issues, has updated the working document on NTSs, issued initially in 2013 and updated in 2022, which includes guidance on a common approach with a view to harmonizing the completion of NTSs within the EU and introducing their update with the results of retrospective project assessments as a feedback and transparency tool [15]. It remains to be seen after the use of this updated document by the scientific community and the Competent Authorities what issues still remain regarding the quality, clarity, and simplicity of NTSs in order to amend future legislation accordingly.

### 2.7. National Reference Laboratories for the Development and Validation of Alternative Methods

National Reference Laboratories (NRLs) for the development and validation of alternative methods as suggested in Article 47 (2) of the Directive 2010/63/EU [1] never managed to be established in some Member States. This indicates a lack of appropriate experts on alternative methods. A proposed solution could be to eventually allow official flexibility for Member States to make collaborations between their laboratories or to clarify that they should not be mandatory, since this kind of laboratory needs resources and expert people trained in alternative methods, of which there is evident shortage in some countries. It should be taken into account that some Member States that have tried to establish a NRL and have not yet succeeded, in addition to a lack of experts, are often hindered by the fact that other pressing priorities are promoted and funded by the Authorities.

### 2.8. Harmonization of the Work of National Committees 

There is guidance developed by the European Commission for all those involved in the oversight of and in the care and use of animals used in scientific procedures, to meet the requirements of the Directive with regard to the composition and duties of establishment Animal Welfare Bodies and the National Committee, which leave flexibility to the Member States [12]. However, this flexibility also leads to a lack of harmonization of the output of various National Committees among Member States. Some of them issue brief recommendations, while others issue booklets of recommendations and guidelines, to aid user establishment Animal Welfare Bodies and project evaluation committees in their daily work. A proposed solution could be the future harmonization of the format of outputs of National Committees without being restrictive.

### 2.9. Statistical Reporting

The European Commission has been publishing statistical reports regarding the use of animals for scientific purposes in Member States since 1991. The reports were published every five years and then every three years from Member States [16]. These reports about animal use for scientific purposes in Member States are now published annually on a European Commission “central database which assists in releasing statistical information on a yearly basis in order to inform and convince the public on the necessity of, and, at the same time, rational use of animals for scientific purposes” [17]. The quality of these statistical reports depends very much upon end users reporting accurate data to their respective Competent Authorities. The former are fully informed of what is required to be reported and under a specific template, which was updated in 2021, in order to include even more detailed significant information and for more animal species, which are required by the current legislative documents [1,18]. Reporting, however, is very complicated, even with these guidelines. There is a risk of creating unreliable statistical data, especially since this work is required to be performed annually, adding much administrative burden to users and Competent Authorities. The proposed solution would be to produce a more simplified and user-friendly template, a need guided by the fact that after many years of implementation, errors still occur due to misunderstandings about the template and its guidelines for use, leading to possibly unreliable statistical data.

### 2.10. Narratives

Narratives are drafted by the Member States’ Central Competent Authorities and are sent annually to the European Commission, which has published them separately for each Member State since 2018, in Part C of the Commission Staff Working Documents on Summary Report on the statistics on the use of animals for scientific purposes [16]. The narrative has a specific template that mainly aims to highlight changes in trends in animal use in the Member State compared to previous years. Significant discrepancies have been noted, though, among the form of explanations offered between Member States’ Competent Authorities.

For instance, in many cases, significant increases or decreases in the use of animals are not sufficiently justified as remarkable trends by the Central Competent Authorities of Member States. Exact numbers of the species used, the re-use of some animals, and the use of Genetically Altered (GA) animals are presented repetitively by most Member States, although these are already presented in EU statistical reports on the use of animals for scientific purposes [16]. The trends in their use, often expressed in percentages, is required and indeed presented by some Member States, who, nevertheless, still do not explain the reasons why these occur or do not mention the number of animals that are involved. For example, a 30% increase in the use of an animal species in a Member State whose users work with a small number of animals has a different impact compared to a 30% increase in use in a Member State that uses a very large number of animals of this species. In addition, some Member States present activity related to specific animal species. All these examples clearly show the need for a harmonized approach in the way justification of animal use is presented by Member States [16].

Another reason for an increase in project authorizations and, relatively, the use of animals, could be large-scale scientific project funding, such as that of Horizon 2020. In certain occasions, this is not often explained by the Competent Authorities of Member States with low numbers of animals used, despite the fact that this declaration is not required by legislation. We suggest that these Competent Authorities could try to offer a general overview of this fact, always taking into account EU data protection and national legislation, plus sensitive data on individual funding of scientists. The aim of this observation is to try to show trends in general funding that significantly increases the number of animals used during some particular years, and not presentation of specific examples that would add unnecessary administrative burden.

### 2.11. Annex III Amendments

Annex III (“Requirements for establishments and for the care and accommodation of animals”) amendments of the Directive 2010/63/EU [1] have proved to be a very slow process. Although there was a provision in Article 50 for this Annex to be modified in a flexible way (and this has been transposed to Member States’ national legislation with relative legislative facilitations) this has not been accomplished so far. For species that are currently not included in the scope of the Directive, each Member State has only the ability to set national rules while awaiting a revision of this Annex. Therefore, the solution currently available for housing requirements of new animal species remains within the Member States without ensuring a harmonization of these requirements. A more flexible and rapid administrative process within the European Union should be considered by legislative officials.

### 2.12. Severity Classification

The severity classification of procedures is very well described in the Directive (Annex VIII) [1] and includes assignment criteria with many examples that assist users in classifying their procedures. Additionally, the European Commission has published a guidance document with illustrative examples that elicit critical thinking in assessing prospective and actual severity of procedures [19]. Furthermore, many relevant educational courses, workshops on severity assessment, e-courses [10], and publications in peer-reviewed journals are available. However, in spite of all these educational tools, which have been in place for many years, several Competent Authorities still depict as discrepancies not only the identification of the severity level of each procedure (e.g., mild procedures that end in the termination of the life of an animal may be declared by users as “non-recovery”), but also the fact that some users do not realize that the overall level of actual severity experienced by an animal in its life cycle is the one that needs to be reported and determines its fate, despite the clear instructions already in place. These discrepancies in applying the severity classification indicate that there is no equal understanding of severity assessments across authorities, and the existing tools and guidance offered by the European Commission are not used thoroughly. We therefore propose that workshops and training on severity assessment are made mandatory for Competent Authorities in order to harmonize severity assessment across Member States. Furthermore, another proposed solution could be the use of standardized templates (such as those included in Annex VII of the Directive 2010/63/EU, e.g., score sheets) which may improve the situation.

### 2.13. Reporting Genetically Altered (GA) Animals

There was no clear provision in the Directive 2010/63/EU for GA animals. The European Commission has tried to address the above issue, plus other important issues regarding GA animals, with updated detailed guidelines, which have indeed proven to be very helpful to users and Competent Authorities [20]. These guidelines are particularly important, as statistical data are now required for each use of GA animals, regardless of their phenotype status and actual severity recorded. The creation of GA and certain breeding phenotype details of GA are also required to be reported [20]. The implementation review of the Directive, according to Article 54(1), which is published every 5 years, requires, among other issues, the reporting of surplus GA animals that are not used for scientific purposes [21]. Reporting of GA and also of all other categories of surplus animals, which are euthanized without having undergone a procedure according to the Directive, is an important matter regarding the Reduction principle. We therefore suggest that future reporting includes all these categories regardless of their procedural status, which can indicate the overall animal use of an establishment and, by extension, the Member State use. We realize that this extended reporting may seemingly produce an increase in the number of animals used in Member States but it will be more representative of the real situation.

### 2.14. Projects or Not?

During these years of the implementation of Directive 2010/63/EU, Central Competent Authorities of Member States have addressed questions originating from tentative project leaders or user establishments’ supervisors about scientific ideas that need to be defined in terms of whether they fall into the scope of the Directive. Some specific examples to illustrate these issues are presented below.

Projects using farmed fish (e.g., European sea bass and gilthead sea bream), both in user establishments and commercial fish farms that have been authorized as such, have a nutritional aim and might not always be considered as projects under the scope of the Directive. It remains unclear if, for instance, changing the diet of fish is considered a procedure under the scope of the Directive 2010/63/EU, and thus needs to undergo a project evaluation and authorization. Several of these nutritional studies may be considered to belong to a grey zone regarding their licensing and reporting. These cases need to be clarified in a potential Revision of the Directive. It is noteworthy that some peer-reviewed journals require a project authorization license to have been issued for these studies, in order to be considered for publication, whereas this requirement is not clear in the Directive, despite the fact that these cases could be considered as procedures according to article 5.b.(ii) [1]. In future legislative proposals, these categories of research should be considered accordingly and be clearly included or excluded.

Central Competent Authorities of Member States often receive queries about whether a research idea is considered a clinical trial or a basic research project under the scope of Directive 2010/63/EU. Clinical trials do not fall under the scope of this Directive. It often remains unclear which studies should be considered as projects and therefore need an authorization, and which do not. For example, there is a dilemma if the application of an authorized medicine (e.g., antibiotics) to farm animals is considered a procedure under the Directive or a veterinary clinical trial. The latter usually requires authorization by the respective national medicines’ agencies of Member States. However, no relevant clear rules are presented in the current text of the Directive 2010/63/EU [1]. It is also noted that farmers are usually reluctant to allow animal research to be conducted on their commercial farm, which has the aim of maximum production and return. Therefore, it is crucial for the scientific community to convince them that the welfare of farm animals is safeguarded when projects take place on site, rather than transporting them to a user establishment for this purpose.

## 3. Conclusions

The European legislative bodies and the scientific community, as years pass, have acknowledged the need for additional guidance to be issued on topics that are not clearly addressed in the available legislative text. The aforementioned issues are only examples that have been identified by the authors that present a need of specific legislative guidance and clarifications, at least in some Member States, in order to maintain animal welfare to the highest standards and provide valuable advice to the user community for the conduct of their duties. One of these is the reporting of non-Technical Project Summaries and their related retrospective assessments, which needs to be conducted in a uniform manner throughout the Member States, including substantial information about the overall animal experience within each project.

Another issue noted is the reporting of surplus animals that are euthanized without having undergone a procedure according to the Directive, which is an important matter regarding the Reduction principle applied by the management of breeder and user establishments. 

The experience of Central Competent Authorities of the Member States of the need for project authorizations of various scientific ideas that might not necessarily fall within the scope of the Directive 2010/63/EU but remain vague in the current text, should be effectively used by European legislative bodies, in order to further contribute to the justified use of animals for scientific purposes in the EU.

It is to be hoped that all actors involved in the protection of animals used for scientific purposes will contribute with their expertise to promote new legislative and guidance documents that will address these and other issues, which will enhance animal welfare.

## Data Availability

No new data were created.

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
