# Peer review of "European Union Legislation for the Welfare of Animals Used for Scientific Purposes: Areas Identified for Further Discussion"

_animals, 2023, doi:10.3390/ani13142367_

Round 1
Reviewer 1 Report
See attached file for general and detailed comments.

Okay, I suppose. But I'm not a native speaker, so maybe have native speaker give it a read to be on the save side?
Author Response
We sincerely thank Reviewer 1 for reviewing our manuscript very carefully and making significant observations that will assist us in clarifying the points identified and providing more substantial and feasible suggestions for future solutions and legislative revisions. We have proceeded to revise many passages according to the Reviewer's comments.
Our replies to each comment in the attached file, and in the manuscript text, are in red font.
The content in the attachment:
Replies to comments of Reviewer 1
General:
The manuscript highlights open issues when it comes to the application of legislature set out by the EU Directive 2010/63/EU (and additional legislative and guidance documents), which may be addressed in a future revision of the Directive. Pointing out and discussing potential shortcomings of legislation when it comes to the transposition of the legislation into practical day-to-day use is an important feedback mechanism for policy makers to prevent practical issues with a legislative framework. Contribution to this discussion –as provided by this manuscript- are important and greatly welcomed.
However, I find that the manuscript as it stands now lacks of clarity and substance to really have a positive impact to this discussion. Most of the issues raised are somewhat known shortcomings and it would need some more substantial suggestions and new ideas to have a positive impact, which brings the issues (and suggestions on how to solve them) forward. On the other hand, I find that some demands lack a realistic assessment on how actual feasible these demands are.
If the authors can re-work some passages accordingly, I think this can be a valuable contribution to the discussion about the legislative framework defined by the EU Directive 2010/63/EU and possible future revisions.
We sincerely thank Reviewer 1 for reviewing our manuscript very carefully and making significant observations that will assist us in clarifying the points identified and providing more substantial and feasible suggestions for future solutions and legislative revisions. We have proceeded to revise many passages according to the Reviewer's comments.
Our replies to each comment below and in the manuscript text are in red font. When the line numbering has changed due to the revision, the new line numbers are mentioned.
Introduction: We have added additional examples of positive aspects of the new Directive in lines 41-42. We have revised a verb in line 59.
Detail:
- 53-56 & l. 293: You say that your aim is to highlight issues and that they may also be addressed in future revisions of the Directive 2010/63. But the phrasing and content for a lot of your highlighted issues is very insubstantial and does not give enough substance to make it clear what you actually want that should be happening. It’s often not really clear how you would like to have specific issues be addressed (in a revision of the Directive) and what you hope will be the positive impact of that. Sometimes it’s not even clear what the issue actually is and you’re mostly describing just a status quo without really highlighting where the problems lay.
This applies specifically for:
- E) Quality of the results of Retrospective Assessment. The text has been revised to be more specific with the addition of clarifications and additional personal suggestions in lines 173-182.
- F) Quality of non-technical project summaries (NTS). The text has been revised with a comment on the potential benefits of applying the updated EC guidance in lines 199-202.
- M) Reporting Genetically Altered (GA) Animals. The text has been revised to be more specific with the addition of clarifications and additional personal suggestions in lines 317-321.
I would very much welcome it if you’d be more clear, direct and bold in describing the issues at hand and your demands, as well as giving constructive suggestions, which can realistically move the discussion forward. I find it very important that impactful and constructive suggestions on how to solve issues are provided in a publication, which wants to highlight issues for future revisions of the Directive. Otherwise it boils down to it being just a list of complaints with no input to improve the situation and I don’t find this an interesting topic for a publication.
We initially tried to highlight problematic issues in a tactful way, so as not to appear critical to the European Parliament and Commission. After receiving your comments, we have made our statements stronger.
-------
- 47-49: It would be good to summarize what the Regulation 2019/1010/EU entails for all readers who are not that familiar with the legislative documents published by the EU.
Thank you very much for the comment. We have added a clarification regarding the provisions of Regulation 2019/1010/EU, in current lines 49-53.
- 49-52: Maybe cite a few examples here for the readers who are not familiar with the documents published by the EU.
We have added here the general “Animals in science” website of the European Commission [4] where all the Guidance documents are available. Further down in the sections, the direct links to each Guidance document are provided.
- 58-60: I’d find it very important to further explain the motivation of your selection of possible improvements here. Why did you chose these topics and not others? Giving your motivation is important for readers to judge the relevance and urgency of these selected issues.
Thank you for this comment. We have now explained our selection has been based on our experience with the implementation of the Directive in our country - Greece - in lines 60, 65-66.
- 64 ff.: You’re describing the status quo and hint on some of the issues that may come with it, but it’s not clear what you actually want to be done about it. Please state clearly what the issue is and your suggestion for a solution.
We are unable to clearly understand what the Reviewer wished to be improved in line 64 ff. We would highly appreciate a further explanation.
l.86: Personally, I would add “..to train official inspectors and authorities…” but that is up to you.
Thank you for indicating this need for clarification. We have clarified that we meant the official inspectors that work for the Competent Authorities and conduct audits of establishments, in lines 91-92.
l.94 ff.: Animals are used for education and training in a wide variety of different areas, hence I would indicate here that the training of veterinary students and the accreditation requirements by one organisation is just one example out of many. Maybe do some research/evaluation if the issues you describe here are generalizable to other areas where animals are used for education and training.
Thank you for this comment. We have indicated that the training of Veterinarians is one example (line 102) and have added aspects of training of other scientists in lines 115-123 (also requested by Reviewer 2, in blue).
l.110: What training needs do you want to have better defined specifically? As mentioned above, animals are used in very different areas and fields, which all may have different requirements for their education and training goals/outcomes, so there needs to be flexibility in the definitions to accustom the different educational needs. So what kind of definition do you have in mind that may be implemented in the Directive 2010/63 or guidance documents?
As an example, we have now suggested the tier-level approach of training in lines 117-122 with an additional reference [9].
- 114 ff.: Please phrase it more clearly and direct what the problem is if certain modules for project evaluators are not provided in member states.
We have now elaborated on the benefits that the provision of a project evaluators’ course with a common training content in all Member States would guarantee a harmonized education and training of member of project evaluation committees, a harmonized approach to project applications, and also in free movement of this personnel in the European Union, in lines 138-142.
- 132: It’s not immediately clear what “membership” does refer to. Membership of what? Would be good to further clarify this.
Thank you for indicating this lack of clarity. We have now specified we mean the composition of Animal Welfare Bodies' members in lines 152 and 154.
- 130 ff.: Please phrase it more clearly and directly what your proposed solution is. Do you want the position of the designated veterinarian be made mandatory for the AWB and that this is included in the Directive 2010/63?
Yes indeed, that was our point. We have added a clarification in lines 156-157 of our opinion that this participation of the designated Veterinarian in the AWB should be mandatory in a future legislative revision.
- 146-148: This seems very specific, but from the way you phrase it here also as if it would be the only problem that the authorities face when collection the needed data. Which I think is not the case. Is this maybe a problem specific for your authorities? If you want to keep this example either ensure that this is a problem that many authorities face or indicate clearly that this is just one example issue that authorities may have to deal with.
We have revised this section in lines 168-182, indicating that occasional difficulties may be noted in some cases in some countries when collecting data on actual severity and harm benefit analysis. We have had this experience in our country, but thought best not to specify, as this may also be the case in other countries. Similarly, as work overload and lack of personnel is a problem in our country, we have added this example as a potential issue. We have also added specific suggestions for a future legislative revision which we believe will be beneficial in lines 174-182.
- 140 ff.: You said your aim is to highlight “grey zones” and how to address them in possible revisions, but here you’re just stating the status quo and it’s not clear what the issue is or what you want to be changed. Please clarify.
The suggestions how the improvement of the quality of retrospective assessments in lines 174-182 can be achieved in a future legislative revision, we believe is now clarified.
l.175: Collaborations between who? Please clarify.
We have revised current line 209 to clarify we mean collaborations between the National Reference Laboratories of the Member States.
- 192: I’m not sure what you mean with “every three years from member states”. Which reports specifically?
We have now clarified that statistical reports were published every 3 years from Member States, and that they are now published annually by the European Commission on the central database, lines 229-232, with the respective additional website [17].
- 199-203: This is an important point. The collection of the data for the statistics is an issue every year again, which may lead to unreliable statistics. Given how much weight is put on these statistics, it’s important they are correct. Hence, I wouldn’t mind if you’d put more emphasize on the importance of this matter.
We agree with your views and have adopted them, indicating the risks of unreliable statistical data in lines 239-241 and 244.
l.213 – 229: I don’t think the demands (to give very specific reasons why numbers of animals used increased/decreased by members states in the annual summary narrative) are feasible. It would require each authority that oversees the application and evaluation of project applications to collect additional data on each individual project application in their country on aspects like specific research contexts and specific funding (which isn’t required yet). This might be particular problematic for member states who have more than one authority who oversees application/evaluation of projects and/or have a high number of project applications per year. Furthermore, it may inflict on data protection laws in some countries to collect data about specific funding of individual scientists. All in all, it would be a huge bureaucratic effort to additionally collect this type of specific statistical data that you demand. And since the reported data already gives information about which areas used how many animals, I don’t see that the data you demand would give such substantial added value that it would justify the vastly increased effort. I agree that the layout and format of the narratives can be harmonized better, but I don’t think it’s feasible to demand this kind of in-depth information and analysis from the authorities and I don’t see the added value. Hence, I would delete or re-work this paragraph.
We appreciate the Reviewer's point of view on the bureaucratic effort to collect this data, and the doubt of its value. Because we believe this animal use is important to be known, we have added a paragraph which we hope describes its importance in lines 268-277.
- 231: For readers not familiar with Annex III please describe what it entails.
Thank you for this suggestion, we have defined it in line 279-280.
- 231 ff.: Again, it’s not really clear what you want to happen. Please clarify.
Thank you for this suggestion, we have clarified with a specific suggestion, lines 287-289.
- 239 ff.: I’m really not sure what you want to happen here. As you stated, the EU (and other organisation) have provided guidance documents, workshops etc. to clarify and harmonize the severity assessment as much as possible, but the fact of the matter is that it is extremely difficult to harmonize when it comes to the practical application and conduction of severity assessments. I don’t know what you have in mind what can be further done (by legislation, the EU, via the EU Directive)? More guidance documents? More workshops? A list, which gives every experiment a defined severity scoring applicable to every single instance that experiment is conducted? I understand and agree that it is frustrating that the severity judgement can differ between projects, but this is due to the severity classification being designed as a case-by-case judgement; the flexibility this entails can lead to a lack of harmonization, sure, but the flexibility is needed to account for differences between labs, environments, handlings, application of 3Rs etc. etc. So I don’t see what can be a realistically achievable improvement here; especially not in regards to legislation. However, I would be interested for any kind of new idea that could improve the situation realistically. So if you have some new and interesting suggestions in that regard, please bring them forward. Generally, please re-work this paragraph accordingly.
Thank you very much for your comment. We have elaborated on the text and added a suggestion for improvement of the current EU legislation, in lines 299-304.
- 265 ff.: This whole paragraph seems very specific. I agree that the discussing about what constitutes an animal experiment (and falls under legislation) can be tricky sometimes and good to point out, but I would approach this issue from a more general point of view and not focus not too much on specific examples. I advice to start with a general view on the issue as introduction and later give a few specific examples (like the farmed fish or farm animals).
Thank you very much for this advice. We have started with an introduction (lines 323-327) on the queries received by the Competent Authorities if scientific ideas fall into the scope of the Directive, and continue with our examples, clarifying them and adding specific suggestions for future legislative revision.
- 276: What exactly is meant with “field trial”?
We meant trials on a farm/field and have revised to "clinical" trial.
l.290 ff.: Please update the conclusion accordingly after changes to the main text have been implemented.
We have revised the Conclusions according to the additions and clarifications made, and have added a paragraph advising the use of the valuable experience of the Competent Authorities of the Member States by future legislative revisions in lines 370-374.
Reference 13: This is just the link to the general “Animals in Science” website of the EU, which holds all the reports, guidance documents etc. This makes it very difficult to find the documents you cite in the text (l. 208, l.218, l.229 etc.). Please don’t just cite this website, but cite the documents more clearly and separately with their own URLs where applicable.
Thank you for this observation. We have placed the general “Animals in Science” website as Reference [4] as mentioned above, and checked/revised the specific links of the other References.
Reviewer 2 Report
This is an interesting piece and highlights some of the main changes the Directive introduced and how it has been developing. As it is a communication, it does not need to be very detailed, but I think the below may help to make it a stronger piece.
· The tense feels a bit off in places. For example, at line 36, you say the Directive ‘was’ in place, which indicates it no longer is.
· The outline of the legislation is good, but there could be a bit more literature and discussion. For example, how has Regulation 2009/1010/EU increased transparency? Some would argue it has not. Whether or not it has been significant will depend on the country.
· I think slightly more detail on some of the recommendations and highlights to specific countries who are good examples would be beneficial.
· When discussing education and training needs, how would you define them?
· There are some claims made that may not be applicable to all member states. It sometimes read as though you have looked at processes and procedures in all member states, and it needs to be clear how this has been done.
· The quality of results of retrospective assessment has been heavily criticised by some in the field, such as the waiver of non-technical summaries under Article 37(2). I do agree that the Directive brought in some benefits, but it isn’t always as strong as the authors are claiming.
· For National Reference Laboratories, how do we know there is an evident shortage? I think it is bold to say that because some member states have struggled to stablish a NRL, we should just do away with them. The importance of developing alternatives cannot just be stated, it needs to be mandated. If some member states have struggled to establish them, they should be encouraging more to join the field through funding means, etc.
Author Response
We sincerely thank Reviewer 2 for the kind words and intention to improve our manuscript. Our replies to each comment in the attached file, and in the manuscript text, are in blue font.
The content in the attachment:
Replies to comments of Reviewer 2
Comments and Suggestions for Authors
This is an interesting piece and highlights some of the main changes the Directive introduced and how it has been developing. As it is a communication, it does not need to be very detailed, but I think the below may help to make it a stronger piece.
We thank Reviewer 2 for the kind words and intention to improve our manuscript. Our replies to each comment below and in the manuscript text are in blue font.
- The tense feels a bit off in places. For example, at line 36, you say the Directive "was" in place, which indicates it no longer is.
Yes indeed, the Directive 86/609/EC mentioned in lines 35-36 was in place from 1986 until the new Directive 2010/63/EU came into force in 2010, and since then no longer is.
- The outline of the legislation is good, but there could be a bit more literature and discussion. For example, how has Regulation 2009/101/EU increased transparency? Some would argue it has not. Whether or not is has been significant will depend on the country.
Thank you very much for the comment. We agree with this observation and have amended accordingly regarding the provisions of Regulation 2019/1010/EU in lines 49-53 and lines 186-187.
- I think slightly more detail on some of the recommendations and highlights to specific countries who are good examples would be beneficial.
It would indeed be beneficial to readers, however we have chosen not to mention specific countries either as good or bad examples in order to avoid provoking any discriminating implications and potentially raise reactions by readers.
- When discussing education and training needs, how would you define them?
This part has been elaborated on, defining in more detail the acquisition of skills that may be necessary for some professions such as Veterinarians or scientists conducting certain procedures, and giving an example of a tier-level approach that could be required by a revision of the legislation in lines 116-123.
- There are some claims made that may not be applicable to all member states. It sometimes read as though you have looked at processes and procedures in all member states, and it needs to be clear how this has been done.
Thank you for this observation. We have tried to address this by providing more information when talking about differences between Member States without referring to specific MSs. For example, in section C) Project Evaluation, lines 135-138, we have added two websites [10,11] that indicate courses conducted in the Member States.
- The quality of results of retrospective assessment has been heavily criticised by some in the field, such as the waiver of non-technical summaries under Article 37(2). I do agree that the Directive brought in some benefits, but it isn't always as strong as the authors are claiming.
We have revised the section E) Quality of the results of Retrospective Assessment, with the addition of clarifications and additional personal suggestions in lines 174-182, as well as the section F) Quality of non-technical project summaries, with a personal observation on the potential benefits of applying the updated EC guidance in lines 199-202.
- For National Reference Laboratories, how do we know there is an evident shortage? I think it is bold to say that because some member states have struggled to establish a NRL, we should just do away with them. The importance of developing alternatives cannot just be stated, it needs to be mandated. If some member states have struggles to establish them, they should be encouraging mores to join the field through funding means, etc.
Thank you very much for your comment, with which we agree. This has now been addressed with the addition of lines 210-213.
Round 2
Reviewer 1 Report
I have a few additional minor points. It would be good if you'd address them for the final manuscript.
Thanks.

Stumbled over one unclear sentence, so maybe just double check the parts of the manuscript you added after the review if everything is correct.
Author Response
Please see the attachment.
The content in the attachment:
Replies to comments of Reviewer 1 – Revision 2
I have a few minor additional comments, which I added in green to the replies.
Dear Reviewer 1,
We sincerely thank you for the new constructive comments which we deeply appreciate, as they continue to help us improve our manuscript significantly.
Our replies to these new comments are added in light blue. In the revised manuscript in addition to blue they are highlighted in yellow.
I would very much welcome it if you’d be more clear, direct and bold in describing the issues at hand and your demands, as well as giving constructive suggestions, which can realistically move the discussion forward. I find it very important that impactful and constructive suggestions on how to solve issues are provided in a publication, which wants to highlight issues for future revisions of the Directive. Otherwise it boils down to it being just a list of complaints with no input to improve the situation and I don’t find this an interesting topic for a publication.
We initially tried to highlight problematic issues in a tactful way, so as not to appear critical to the European Parliament and Commission. After receiving your comments, we have made our statements stronger.
I think it’s fine to bring forward issues you observed and which may benefit from improvements in a professional manner, which you did. The EU is a democratic institution; it is supposed to get critical feedback from EU citizens. Otherwise how would they know what people want them to improve? You can maybe add a few words about your suggestion are not intended to be read as accusations or overly critical, but aiming very much on trying be beneficial and having a positive impact on the application of the Directive and animal welfare, if you worry that you may come across as confrontational.
We thank Reviewer 1 for the expression of these thoughts, with which we agree. We adopted the Reviewer’s wording suggestion of our trying to improve the situation without being overly critical in lines 61-63.
-------
- 58-60: I’d find it very important to further explain the motivation of your selection of possible improvements here. Why did you chose these topics and not others? Giving your motivation is important for readers to judge the relevance and urgency of these selected issues.
Thank you for this comment. We have now explained our selection has been based on our experience with the implementation of the Directive in our country - Greece - in lines 60, 65-66.
You could maybe add something along the lines that while your experiences with the Directive derives from working in one member state –Greece- you think a lot of these issues are applicable to other member states as well; maybe not one-to-one, but to certain degrees for sure. But that is up to you.
We thank Reviewer 1 again for this very kind suggestion for elaborating on the potential applicability to other Member States as well, and have adopted this suggestion in lines 69-71.
--------------------------
- 64 ff.: You’re describing the status quo and hint on some of the issues that may come with it, but it’s not clear what you actually want to be done about it. Please state clearly what the issue is and your suggestion for a solution.
We are unable to clearly understand what the Reviewer wished to be improved in line 64 ff. We would highly appreciate a further explanation.
In l. 70-89 (edited manuscript) you describe the status quo and some aspects which are unclear (such as the needed qualifications to enrol in some courses), but you a) don’t really explain why this is a problem and what consequences may arise from this lack of clarification and b) you don’t give any suggestions how these unclarified issues may be solved. Right now the paragraph reads like sure, there’re some aspects which are a bit unclear and up for interpretation, but it doesn’t seem like these aspects cause any harm, so there is no need to change anything.
Thank you very much for this observation. We have now clarified that we mean that because of the vague wording of the Framework document, there may be differences between Member States in the transposition of the Directive regarding Function B persons’ E&T starting qualifications. We now specify that these potential differences can hinder harmonization and free movement of Function B persons. We now suggest this should be addressed with clarifications in a future legislative revision. Within lines 89-100.
-----------------------------------------------------
l.213 – 229: I don’t think the demands (to give very specific reasons why numbers of animals used increased/decreased by members states in the annual summary narrative) are feasible. It would require each authority that oversees the application and evaluation of project applications to collect additional data on each individual project application in their country on aspects like specific research contexts and specific funding (which isn’t required yet). This might be particular problematic for member states who have more than one authority who oversees application/evaluation of projects and/or have a high number of project applications per year. Furthermore, it may inflict on data protection laws in some countries to collect data about specific funding of individual scientists. All in all, it would be a huge bureaucratic effort to additionally collect this type of specific statistical data that you demand. And since the reported data already gives information about which areas used how many animals, I don’t see that the data you demand would give such substantial added value that it would justify the vastly increased effort. I agree that the layout and format of the narratives can be harmonized better, but I don’t think it’s feasible to demand this kind of in-depth information and analysis from the authorities and I don’t see the added value. Hence, I would delete or re-work this paragraph.
We appreciate the Reviewer's point of view on the bureaucratic effort to collect this data, and the doubt of its value. Because we believe this animal use is important to be known, we have added a paragraph which we hope describes its importance in lines 268-277.
I don’t quite understand the sentence l. 269-272 (“…not often noted,…..despite the fact that this is not required…”). Probably just a typo? Otherwise maybe rephrase a bit to clarify.
Thank you very much for your comment. “Noted” has been replaced by “explained” hoping to make the phrase clear.
----------------------------------
- 239 ff.: I’m really not sure what you want to happen here. As you stated, the EU (and other organisation) have provided guidance documents, workshops etc. to clarify and harmonize the severity assessment as much as possible, but the fact of the matter is that it is extremely difficult to harmonize when it comes to the practical application and conduction of severity assessments. I don’t know what you have in mind what can be further done (by legislation, the EU, via the EU Directive)? More guidance documents? More workshops? A list, which gives every experiment a defined severity scoring applicable to every single instance that experiment is conducted? I understand and agree that it is frustrating that the severity judgement can differ between projects, but this is due to the severity classification being designed as a case-by-case judgement; the flexibility this entails can lead to a lack of harmonization, sure, but the flexibility is needed to account for differences between labs, environments, handlings, application of 3Rs etc. etc. So I don’t see what can be a realistically achievable improvement here; especially not in regards to legislation. However, I would be interested for any kind of new idea that could improve the situation realistically. So if you have some new and interesting suggestions in that regard, please bring them forward. Generally, please re-work this paragraph accordingly.
Thank you very much for your comment. We have elaborated on the text and added a suggestion for improvement of the current EU legislation, in lines 299-304.
The changes are fine. But if I may throw in a suggestion for the last sentence of that paragraph (feel free to either use it, rephrase it or reject it): “…that the overall level of actual severity experienced by an animal in its life cycle is the one that needs to be reported and determines its fate, despite the clear instructions already in place. These discrepancies in applying severity classification indicates that there is no equal understanding of severity assessments across authorities and the tools and guidance offered by the EU Commissions are not used thoroughly. We propose therefore that workshops and training on severity assessment are made mandatory for competent authorities to harmonize severity assessment across member states. Furthermore, the use of standardized templates (such as these included in Annex VII of the Directive 2010/63/EU) may improve the situation.”
We sincerely thank Reviewer 1 for dedicating time to elaborate on this issue so effectively and totally agree with the suggestion for the mandatory training of Competent Authorities. We have taken the liberty to use this suggestion and wording in our revision in lines 316-323.